# A Comparative Pharmacokinetic Study of Fexuprazan 10 mg: Demonstrating Bioequivalence with the Reference Formulation and Evaluating Steady State

**DOI:** 10.3390/ph16081141

**Published:** 2023-08-11

**Authors:** Wonsuk Shin, A-Young Yang, Hyung Park, Hyejung Lee, Hyounggyoon Yoo, Anhye Kim

**Affiliations:** 1Department of Clinical Pharmacology and Therapeutics, CHA Bundang Medical Center, CHA University School of Medicine, Seongnam 13520, Gyeonggi-do, Republic of Korea; wonsug89@chamc.co.kr (W.S.); adud610@gmail.com (A.-Y.Y.); hgyoo0317@cha.ac.kr (H.Y.); 2CHA Global Clinical Trial Center, CHA Bundang Medical Center, Seongnam 13520, Gyeonggi-do, Republic of Korea; 3Clinical Development Center, Daewoong Pharmaceutical Co., Ltd., Seoul 06170, Republic of Korea; 2210467@daewoong.co.kr (H.P.); hyejunglee@daewoong.co.kr (H.L.); 4Institute for Biomedical Informatics, CHA University School of Medicine, CHA University, Seongnam 13488, Gyeonggi-do, Republic of Korea

**Keywords:** pharmacokinetics, bioequivalence, gastritis, multiple dose, fexuprazan

## Abstract

Fexuprazan is a potassium-competitive acid blocker approved for treating gastric-acid-related diseases. Although the effectiveness of the recent formulation fexuprazan 10 mg has been demonstrated in Phase 3 clinical trials, data on the pharmacokinetics (PKs) of administering fexuprazan 10 mg twice daily at a 12 h interval are lacking. Moreover, it is imperative to ensure the bioequivalence of the new formulation with the previously approved 40 mg formulation. This study evaluated the pharmacokinetics (PKs) of the single- and multiple-dose oral administration of fexuprazan 10 mg tablets in healthy participants (Part 1) and investigated their bioequivalence with 40 mg tablets (Part 2). Part 1 comprised a single- and multiple-dose, one-sequence, two-period design and eight participants, while Part 2 comprised a single-dose, 2 × 2 crossover design and 24 participants. In Part 1, in Periods 1 and 2, participants received single and multiple doses (twice daily) of fexuprazan 10 mg, respectively. The maximum plasma concentration (C_max_) area under the concentration–time curve from 0 to 12 h (AUC_0–12h_) of the multiple-dose participants was approximately double that of the single-dose participants. In Part 2, the geometric mean ratios (90% confidence intervals) for C_max_ and AUC from zero to the time of the last quantifiable concentration (AUC_last_) of the use of four fexuprazan 10 mg tablets to those of one fexuprazan 40 mg tablet were 1.0290 (0.9352–1.1321) and 1.0290 (0.9476–1.1174), respectively, meeting the bioequivalence criteria. Favorable PKs were observed after single and multiple administrations of one fexuprazan 10 mg tablet, and four fexuprazan 10 mg tablets were pharmacokinetically equivalent to one fexuprazan 40 mg tablet.

## 1. Introduction

Gastric-acid-related diseases (ARDs), which are primarily caused by the excessive production or imbalance of gastric acid in the stomach, mainly include gastroesophageal reflux disease (GERD), peptic ulcers, gastritis, and Zollinger–Ellison syndrome [1]. Among ARDs, gastritis is one of the most commonly clinically diagnosed diseases worldwide, and its prevalence is gradually increasing in Korea [2]. Gastritis manifests as the inflammation or irritation of the stomach lining and may be caused by many factors, including infection, alcohol, particular medications, and some allergy and immune conditions [3]. Treatments for ARDs are mainly designed to reduce gastric acid production, relieve symptoms, and promote tissue healing. Thus, antacids, such as proton pump inhibitors (PPIs) and H_2_ blockers, have been widely used to treat ARDs [4,5]. Although empirical treatment for ARDs primarily relies on conventional drugs, and the effects of H_2_-receptor antagonists and PPIs on the endoscopic improvement of acute and chronic gastritis have been reported [6], some patients may not respond well to these treatments [7,8,9]. Therefore, more effective medications are required to offer enhanced symptom relief for patients not experiencing sufficient symptom improvement with the existing therapeutic options.

Potassium-competitive acid blockers (P-CABs), a new class of acid suppressants, have recently been developed as an alternative treatment [10]. P-CABs, including vonoprazan, tegoprazan, and fexuprazan, are significantly more efficient than PPIs in suppressing gastric acid secretion [11]. As P-CABs do not require acid activation, they exhibit a faster onset than PPIs [12]. Additionally, they effectively reduce nocturnal acid breakthroughs owing to their longer action duration [13]. P-CABs are predominantly metabolized by cytochrome P450 (CYP) 3A4, which has little inter-individual variability. In contrast, many PPIs are mainly metabolized by CYP2C19 and may subsequently exhibit evident differences in drug exposure. In South Korea, the Ministry of Food and Drug Safety (MFDS) has recently approved fexuprazan, a P-CAB developed by Daewoong Pharmaceutical Co., Ltd. (Seoul, Korea), for use in two formulations of 10 [14] and 40 mg [15]. It is the first P-CAB to be approved for acute and chronic gastritis treatment in South Korea [16]. Fexuprazan 10 mg is a new formulation developed to improve patient medication compliance and market competitiveness by changing the shape of the tablet from round to oblong [17]. Fexuprazan 40 mg administered once daily is appropriate for therapeutic use in treating erosive GERD [18], whereas fexuprazan 10 mg administered twice daily is prescribed for ameliorating gastric mucosal lesions associated with acute and chronic gastritis [19]. Although the effectiveness of fexuprazan 10 mg (twice daily) has been demonstrated in Phase 3 clinical trials on patients with gastritis, data on the pharmacokinetics (PKs) of administering fexuprazan 10 mg twice daily at a 12 h interval are lacking. Therefore, further investigation is necessary to address this crucial information gap. Moreover, it is imperative to ensure the bioequivalence of the new formulation with the previously approved 40 mg formulation.

The present study aimed to evaluate the PKs of the new, recently approved, fexuprazan 10 mg formulation after single and multiple administrations (Part 1), and its bioequivalence to the previously approved 40 mg formulation was investigated (Part 2).

## 2. Results

### 2.1. Demographic Characteristics

Eight male participants were enrolled in and completed Part 1 of the study, and their PKs, safety, and demographics were evaluated. The mean (±standard deviation) age, height, weight, and BMI were 32.63 (±9.74) years, 176.58 (±7.44) cm, 74.76 (±7.50) kg, and 23.95 (±1.55) kg/m^2^, respectively. Twenty-four participants were enrolled in Part 2 and randomly assigned to Sequence A (nine male and three female participants) or B (ten male and two female participants). Of the participants in Part 2, one was excluded because of a failure to follow-up on 3d of Period 2, and 23 participants completed the clinical trial. Accordingly, the PK profiles of the 23 participants who completed the study and the safety profiles of the 24 participants who received at least one dose of IPs were analyzed. The mean (± standard deviation) age, height, weight, and body mass index (BMI) were 27.17 (±6.63) years, 172.24 (±8.50) cm, 72.48 (±10.90) kg, and 24.33 (±2.46) kg/m^2^, respectively. There were no significant differences between the sequence groups in terms of mean age, height, weight, or BMI (*p*-values = 0.1812, 0.7203, 0.4868, and 0.2141, respectively).

### 2.2. PKs of Fexuprazan 10 mg after Single and Multiple Administration (Part 1)

The PK parameters of single and multiple doses of fexuprazan are summarized in Table 1. During Period 1, after a single dose of fexuprazan 10 mg administered once daily, the plasma concentration of fexuprazan peaked at 2.25 h, and the mean maximum plasma concentration (C_max_) and AUC_0–12h_ were 8.26 ng/mL and 53.47 h∙ng/mL, respectively. The plasma fexuprazan concentration showed rapid absorption, declining in a polyphasic manner (Figure 1A).

During Period 2, following multiple doses of fexuprazan 10 mg administered twice daily, the plasma fexuprazan concentration peaked at 2 h, and the mean C_max,ss_, AUC_0–12h,ss_, and C_min,ss_ were 13.85 ng/mL, 109.73 h∙ng/mL, and 5.36 ng/mL, respectively. Following the repeated administration of fexuprazan 10 mg twice daily, its plasma concentrations at 7d 12 h, 8d 0 h, and 8d 12 h were compared to the slope (β_1_ = 0) using a regression equation to determine whether a steady state had been reached. At this point, the concentration gradient (90% CI) was −0.0004 (−0.0168–0.0160), indicating that a steady state had been reached. Multiple doses accumulated twice as much as a single dose, and the calculated mean accumulation ratio was 2.11. After administering multiple doses of fexuprazan 10 mg twice daily, a steady state was achieved. Moreover, rapid absorption was observed up to the peak, followed by a rapid decline in the plasma concentration–time curves for up to 12 h.

### 2.3. PKs of a Single Dose of Four Fexuprazan 10 mg Tablets and a Single Dose of One Fexuprazan 40 mg Tablet (Part 2)

The PK parameters of fexuprazan after a single dose of four fexuprazan 10 mg (T1) tablets and after a single dose of one fexuprazan 40 mg (T2) tablet are summarized in Table 1. An overlap was observed in the mean plasma concentration–time curves between T1 and T2 (Figure 1B). The rapid absorption of T1 and T2 was detected for up to 2.50 and 3.00 h, respectively. After reaching C_max_ (T1, 34.62 ng/mL; T2, 33.87 ng/mL), the plasma concentration–time curves exhibited rapid declines for up to 12 h, followed by slow declines for up to 48 h, indicating a similar half-life for T1 and T2 (T1, 9.91 h; T2, 9.44 h). The mean AUC_last_ and AUC_inf_ values for T1 were 463.19 and 479.88 h∙ng/mL, respectively, and those for T2 were 446.24 and 459.82 h∙ng/mL, respectively. When administered with T1 and T2, the primary PK parameters, C_max_ and AUC_last_, were comparable. Moreover, the point estimates and 90% CIs of the geometric mean ratios (GMRs) (GMR = T1/T2) of C_max_ and AUC_last_ were 1.0290 (0.9352–1.1321) and 1.0290 (0.9476–1.1174), respectively (Table 2). The 90% CIs of the analyzed PK parameters were within the bioequivalence standard range of 0.80–1.25. The intra-subject CV (%), calculated using Equation (1), was <19% for both C_max_ and AUC_last_. There was no noticeable within-subject difference between C_max_ and AUC_last_ for the reference or test formulations (Figure 2).
(1)[CV=100×sqrt(exp(Mean Square Error)−1)]

### 2.4. Safety Assessment

In Part 1, one participant experienced two TEAEs, both of which were headaches. One treatment-emergent adverse event (TEAE) occurred with a single fexuprazan 10 mg dose, whereas the other occurred after multiple fexuprazan 10 mg doses. Two TEAEs were assessed as adverse drug reactions (ADRs) but were mild and resolved over time without any action.

In Part 2, five participants experienced eight TEAEs; three TEAEs were reported by two participants in T1, and five TEAEs were reported by four participants in T2. The details of the TEAEs are summarized in Table 3. The safety analysis revealed no significant differences in TEAE incidence between the two treatment groups (*p* = 0.3827). Seven TEAEs (T1, three TEAEs reported by two participants; T2, four TEAEs reported by three participants) reported by four participants were assessed as ADRs. The reported ADRs included abdominal pain (two TEAEs in two participants), diarrhea (one TEAE in one participant), hematochezia (one TEAE in one participant), nausea (one TEAE in one participant), and headache (one TEAE in one participant). All TEAEs were mild and resolved spontaneously within a few hours or days. No TEAE leading to investigational product (IP) withdrawal, serious adverse events, or death was observed. No clinically significant findings were reported in terms of physical examination, vital signs, or 12-lead electrocardiogram (ECG). Safety data, including TEAEs, laboratory tests, vital signs, and ECG, confirmed that T1 was as safe and tolerable as T2 in healthy adults.

## 3. Discussion

We successfully conducted a two-part study according to the objectives of our research. In Part 1, the safety and PK profiles after single and multiple administrations of fexuprazan 10 mg were evaluated in eight participants. After a single administration, PKs were evaluated for 3 d, with twice daily administration following a 4-day washout period. Considering the observed terminal half-life of approximately 9 h after the single administration, the PKs after multiple administrations were assessed under steady-state conditions. The observed AUC_0–12,ss_ after administering fexuprazan 10 mg at 12 h intervals was 109.73 h∙ng/mL in the present study, corresponding to half of the AUC_0–24,ss_ (223.7 h∙ng/mL) after the once-daily administration of fexuprazan 20 mg that was reported in a first-in-human study [20]. This result suggests that administering fexuprazan 10 mg twice daily and fexuprazan 20 mg once daily result in similar exposure over 24 h. After multiple fexuprazan dose administrations in the first-in-human study, a significant correlation was observed between gastric pH parameters and doses in the range of 20–160 mg [20]. Thus, the comparable values of AUC_0–24,ss_ after administering fexuprazan 10 mg twice daily and fexuprazan 20 mg once daily suggest similar gastric pH parameters, including the percentage of time that the gastric pH was ≥4.0. Our findings report therapeutic effects similar to those observed in the Phase 3 clinical trial conducted on patients with acute and chronic gastritis, indicating that both the once-daily dose of 20 mg and the twice-daily dose of 10 mg exhibited superior efficacy compared to the placebo [19].

In one subgroup analyses, regardless of *Helicobacter pylori* infection, the group receiving fexuprazan 10 mg twice daily showed a significantly higher erosion improvement rate than the placebo group [19]. Specifically, in patients with *H. pylori* infection, members of the group receiving a twice-daily dose of fexuprazan 10 mg showed a significantly higher erosion improvement rate than in those without *H. pylori* infection (81.8% vs. 61.3%, respectively). In addition, in patients with chronic gastritis, the twice-daily fexuprazan 10 mg group showed significantly higher erosion improvement rates than the once-daily fexuprazan 20 mg group (66.7% (10 mg b.i.d.) vs. 43.3% (20 mg q.d.)) [19]. Therefore, these previous findings and ours suggest that fexuprazan can be effectively used for various ARDs based on comparable exposure and efficacy using either a once-daily dose of 20 mg or a twice-daily dosage regimen.

Part 2 was conducted to compare the PK and safety profiles of the two formulations containing fexuprazan 40 mg; The test formulation (four tablets of fexuprazan 10 mg) had PK characteristics bioequivalent to those of the reference formulation (one tablet of fexuprazan 40 mg). This conclusion was supported by the finding that the C_max_ and AUC_last_ GMRs (90% CI) were within the conventional bioequivalence criteria following the administration of each formulation. In addition, the mean plasma concentration–time profiles of the two formulations were superimposable from pre-dosing to 48 h after dosing, and their safety profiles were similar. These results indicate that the test formulation of fexuprazan can be used as an alternative to the reference formulation without significant differences in systemic exposure and safety. Moreover, the sample size and design of this study were appropriate for evaluating the bioequivalence and safety profiles of the two drugs. In this study, to attain an intra-subject variability of C_max_ of 19% for fexuprazan 40 mg, the minimum sample size required was 18 participants. Twenty-four participants were enrolled, and 23 completed the study. The number of participants was sufficient to minimize β-errors (type II), and the randomization of the study groups was sufficiently balanced to avoid bias associated with sequence allocation. The PK sampling time points were well established to observe the T_max_ and systemic exposure, which was supported by the fact that the AUC_extra_ (%) was <5% in both the test and reference formulations.

Among the P-CAB class of drugs, fexuprazan is the only drug approved in South Korea for gastritis. Fexuprazan has a longer half-life than other P-CABs, such as vonoprazan (6.95 h) and tegoprazan (3.65–5.39 h) [21], and in preclinical studies, gastric acid secretion was found to be similar to or greater than that of drugs of the same class [20]. Therefore, efforts are being made to develop different formulations and strengths of fexuprazan to enhance clinical practice, expand indications, and improve medication convenience. The 10 mg formulation used in this study is a newly developed formulation that differs from the formulation used in the first-in-human study [20]; it was changed from a white, round, film-coated tablet weighing 156 mg to an orange, oblong, film-coated tablet weighing 157.5 mg (Appendix A, Appendix A). Oblong-shaped tablets are easier to swallow and have faster transit times in the esophagus than round-shaped tablets of the same weight [17]. Considering that chronic ARDs can cause frequent irritation to the esophagus, leading to complications, such as dysphagia [22,23], formulations of changed shapes might help improve medication compliance in patients with gastritis.

This study had several limitations. First, PKs, safety, and tolerability were evaluated in relatively few healthy volunteers. Nevertheless, the bioequivalent drug exposure and pharmacokinetic–pharmacodynamic (PK/PD) relationships suggest favorable outcomes, effectiveness, and better compliance in a large number of patients with gastritis. Second, this study was conducted only on healthy Korean participants. However, Hwang et al. showed that PK profiles, PK/PD relationships, gastric acid suppression, and safety profiles were similar among Korean, Caucasian, and Japanese participants after the administration of single and multiple doses of fexuprazan [24].

## 4. Materials and Methods

### 4.1. Participants and Study Design

This study was conducted at the Global Clinical Trials Center of the CHA Bundang Medical Center, CHA University, Seongnam, Republic of Korea, with strict adherence to the key ethical principles outlined in the Declaration of Helsinki, the Good Clinical Practice Guidelines of the International Council for Harmonization of Technical Requirements for Pharmaceuticals for Human Use, and local laws and regulations. The study protocol was reviewed and approved by the MFDS and Institutional Review Board (IRB) of CHA University (IRB no. CHAMC 2021-06-021). Furthermore, the study has been registered and can be found on ClinicalTrials.gov (https://clinicaltrials.gov, accessed on 8 December 2021) under the identifier NCT05149274.

Before initiating the clinical trial, the investigators provided all participants with the study information and relevant details. Screening procedures were conducted exclusively for individuals who voluntarily consented to participate in the clinical trial. The inclusion criteria were healthy adults aged 19–45 years, assessed for eligibility based on medical history, physical examination, vital signs, clinical laboratory tests, and a 12-lead ECG. The exclusion criteria included individuals with a clinically significant medical history, those who had participated in another clinical trial within six months preceding the screening, and those taking prescribed medications that could not be temporarily discontinued for at least two weeks before the screening.

This study included two parts (Figure 3). Part 1 followed an open-label, single- or multiple-dose, 1-sequence, and 2-period design and aimed to evaluate the PKs of the single-dose and multiple-dose oral administration of fexuprazan 10 mg tablets in healthy participants. It involved eight participants and aimed to explore their PK characteristics after single- and multiple-dose fexuprazan 10 mg administration. The number of participants was determined based on the ascending dose cohort in the first-in-human study on fexuprazan [20]. The participants were admitted to the clinical trial center a day before the first administration and were instructed to keep fasting state for at least 10 h prior. The next day (1d), at around 9 am, the participants received a single dose of fexuprazan 10 mg (T0) with 150 mL of water, and blood samples (6 mL) were collected at various time points, including pre-dosing (0 h) and post-dosing (after 0.5, 1, 1.5, 2, 2.5, 3, 4, 6, 8, 12, 24, and 48 h). After a 4-day washout period, the participants visited the clinical trial center and were administered multiple doses of fexuprazan 10 mg twice daily (T0′) with 150 mL of water at approximately 9 am and 9 pm for 4 days (D5–7: approximately 9 am and 9 pm; D8: approximately 9 am). Upon reaching a steady state (D8), blood sampling was conducted at the same time points as in Period 1, up to 12 h, to determine the area under the concentration–time curve (AUC) from time zero to the end of the dosing interval.

Part 2 employed a randomized, open-label, single-dose, 2-sequence, 2-period, crossover design and aimed to confirm the bioequivalence between a single dose of four fexuprazan 10 mg tablets (T1) and a single dose of one fexuprazan 40 mg tablet (T2). The participants were randomly assigned to Sequence A or B. Sequence A participants received T1 in Period 1 and T2 in Period 2. Sequence B participants received T2 in Period 1 and T1 in Period 2. Blood samples (6 mL) were collected at pre-dosing (0 h) and post-dosing (0.5, 1, 1.5, 2, 2.5, 3, 4, 6, 8, 12, 24, and 48 h) during each period. According to a previous study on fexuprazan, the intra-subject CVs for C_max_ and AUC from zero to the time of the last quantifiable concentration (AUC_last_) were approximately 19.8% and 16.8%, respectively [12]. To assume an actual (four fexuprazan 10 mg tablets)/(one fexuprazan 40 mg tablet) ratio of 1.1 and an equivalence range of 0.8–1.25, a minimum of 18 participants would be required to achieve a statistical power of at least 80% at a significance level of 0.05. Therefore, considering a dropout rate of 25%, the target sample size was set at 24 participants.

The fasted study participants were administered IPs (either fexuprazan 10 mg or 40 mg) with 150 mL of water. The IPs used in this study were developed by Daewoong Pharmaceutical Co., Ltd. Blood samples were promptly collected in heparinized tubes at each blood sampling time point, and the plasma was separated through centrifugation at 1900× *g* for 10 min at 4 °C. The collected plasma samples were then transferred to 2 Eppendorf tubes at a volume of approximately ≥1.0 mL and stored in a freezer at ≤−70 °C until further analysis.

Participants were admitted the day before the administration of the IPs and discharged the day after dosing. Throughout their hospital stay, all meals were provided, except for breakfast on the day of dosing, and participants were only allowed to consume the meals and beverages provided by the clinical trial center. The decision to participate in the clinical trial was entirely voluntary for the participants, and declining to participate had no consequences. Furthermore, participants retained the right to withdraw from the trial at any point, without facing any disadvantages regarding their future medical care. During the duration of the clinical trial, the researchers prioritized the safety of the participants, taking all necessary precautions to promptly and appropriately minimize any potential adverse effects, if they were to occur. Insurance coverage was obtained to cover any injuries that participants might sustain during the clinical trial procedures.

### 4.2. Determination of Plasma Fexuprazan Concentration

Human plasma was analyzed using validated liquid chromatography–tandem mass spectrometry (LC-MS/MS). The chromatographic separation of fexuprazan and internal standard (IS) was achieved using an ACE C18, 2.1 × 50 mm, and a 3 μm column (Aberdeen, Scotland) under isocratic conditions with a flow rate of 0.5 mL/min. Fexuprazan and IS were detected using an AB SCIEX API 5000 mass spectrometer (Applied Biosystems/MDS Sciex, Foster City, CA, USA) using multiple reaction monitoring in the positive electrospray ionization mode. The mass transitions for fexuprazan and IS were *m*/*z* 411.3 → 380.0 and *m*/*z* 414.3 → 380.2, respectively, and the method was validated over a concentration range of 0.1 to 100 ng/mL. The lower limit of quantitation was 0.1 ng/mL. The within-run precision and accuracy were 1.0 to 5.7% and −7.3 to 5.0%, respectively, and the between-run precision and accuracy were 1.0 to 6.4% and −1.0 to 1.0%, respectively.

### 4.3. PK Assessment

Non-compartmental analysis (NCA) was performed using the Phoenix WinNonlin software version 8.3 (Certara Co., Princeton, NJ, USA) to determine the PK parameters of fexuprazan. In Part 1, the PK parameters were C_max_, C_max_ at steady state (C_max,ss_), the minimum observed concentration at steady state during the dosing interval (C_min,ss_), time to reach C_max_ (T_max_), time to reach C_max,ss_ (T_max,ss_), AUC from 0 to 12 h (AUC_0–12h_), AUC_0–12h_ at steady state (AUC_0–12h,ss_), and AUC_last_. The accumulation ratio based on AUC_0–12h_ was calculated as AUC_0–12,ss_ multiple dose/AUC_0–12_ after a single dose.

In Part 2, the PK parameters included C_max_, T_max_, AUC_last_, the AUC from zero to infinity (AUC_inf_), apparent volume of distribution (V_z_/F), elimination half-life (t_1/2_), and apparent clearance (CL/F); of these, C_max_, C_max,ss_, T_max_, T_max,ss_, and C_min,ss_ were the actual observed values. AUC_0–12h_, AUC_0–12h,ss_, AUC_last_, and AUC_inf_ were calculated using the linear trapezoidal rule. The elimination rate constant (k_e_) was estimated by the linear regression of the terminal declining phase in the logarithmic plasma concentration–time profile, and t_1/2_ was calculated from the ratio of the natural logarithm of 2 to k_e_. AUC_inf_ was determined by adding AUC_last_ to the extrapolated area beyond the last observed concentration (C_last_), AUC_last_ + C_last_/k_e_. CL/F was calculated as Dose/AUC_inf_, and V_z_/F was calculated as (CL/F)/k_e_.

### 4.4. Safety and Tolerability Assessment

Safety was evaluated based on TEAEs, physical examinations, vital signs, 12-lead ECGs, and clinical laboratory tests. TEAEs were either spontaneously reported by participants or identified through data collected during scheduled interviews throughout the study period. All TEAEs were coded according to the Medical Dictionary for Regulatory Activities (MedDRA) version 24.0 and summarized by treatment, severity, and relationships with IPs.

### 4.5. Statistical Analysis

Descriptive statistics were used to summarize baseline demographics, such as age, weight, height, and BMI. The PK parameters and safety profiles were evaluated using descriptive statistics. All statistical analyses were performed using SAS version 9.4 (SAS Institute Inc., Cary, NC, USA). In Part 1, the method used to determine whether the curve is flat involved stepwise testing for linear trends to assess the steady state. This assessment utilized pre-dose samples obtained on the day before the PK assessment day and on the PK assessment day [25]. For the analysis, plasma concentrations at 7d 12 h, 8d 0 h, and 8d 12 h were considered.

In Part 2, the primary PK endpoints (C_max_ and AUC_last_) were log-transformed to develop a mixed-effect model, with treatment effects as fixed effects and participant effects as random effects. GMRs with 90% confidence intervals (Cis) of the primary PK parameters between the two treatment groups (T1 vs. T2) were estimated to evaluate bioequivalence. Bioequivalence for T1 vs. T2 was determined by assessing whether the 90% CI of the GMRs of the primary PK parameters were within the bioequivalence range of 0.8 to 1.25. Numerical data, such as demographic characteristics and PK parameters between the two treatments, were compared using an independent *t*-test or the Wilcoxon rank-sum test. Categorical data, such as TEAE incidence, were compared using the chi-square or Fisher’s exact test.

## 5. Conclusions

The new fexuprazan 10 mg formulation demonstrated favorable safety and tolerability when administered in single or multiple doses and exhibited bioequivalence compared to fexuprazan 40 mg. Therefore, this new formulation can be effectively used in developing dosing regimens for various gastric ARDs, including acute and chronic gastritis.

## Figures and Tables

**Figure 1 pharmaceuticals-16-01141-f001:**
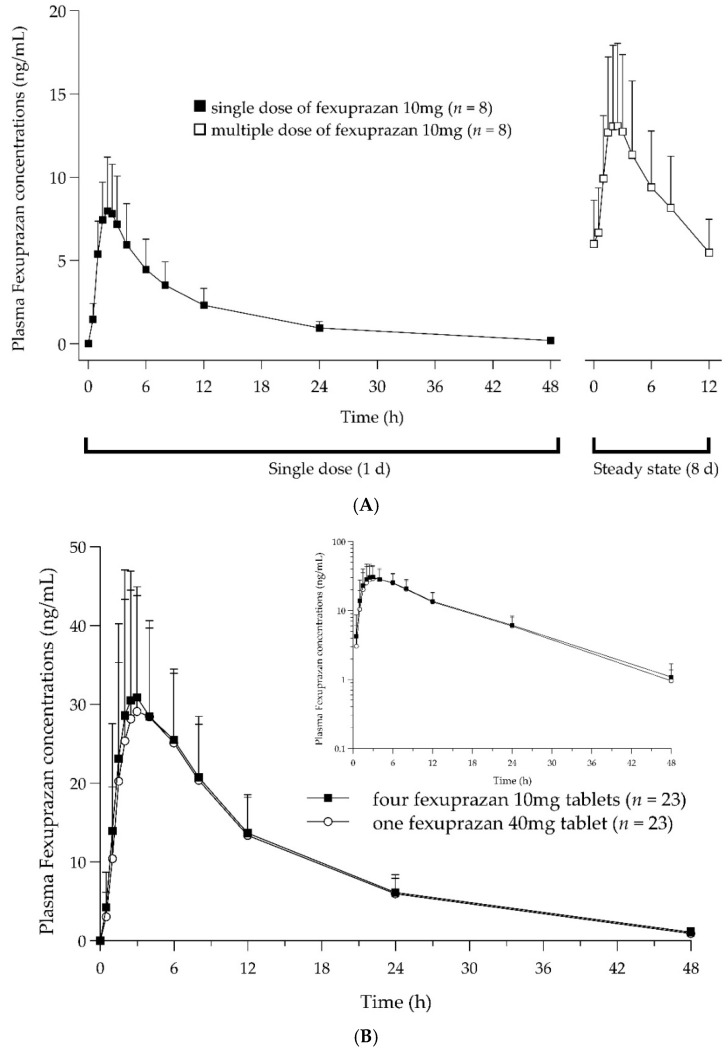
Plasma concentration–time curve. (**A**) After administration of a single dose (black square) and multiple doses (white square) of fexuprazan 10 mg. (**B**) After administration of four fexuprazan 10mg tablets (T1, black square) and one fexuprazan 40 mg tablet (T2, white circle).

**Figure 2 pharmaceuticals-16-01141-f002:**
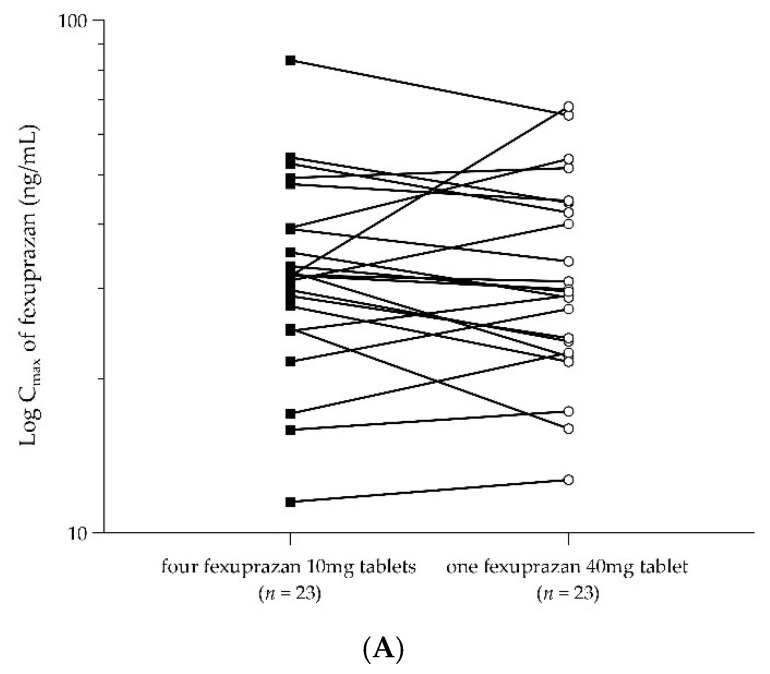
Participant profiles. (**A**) C_max_ and (**B**) AUC_last_ after administration of four fexuprazan 10 mg tablets (T1, black square) or one fexuprazan 40 mg tablet (T2, white circle).

**Figure 3 pharmaceuticals-16-01141-f003:**
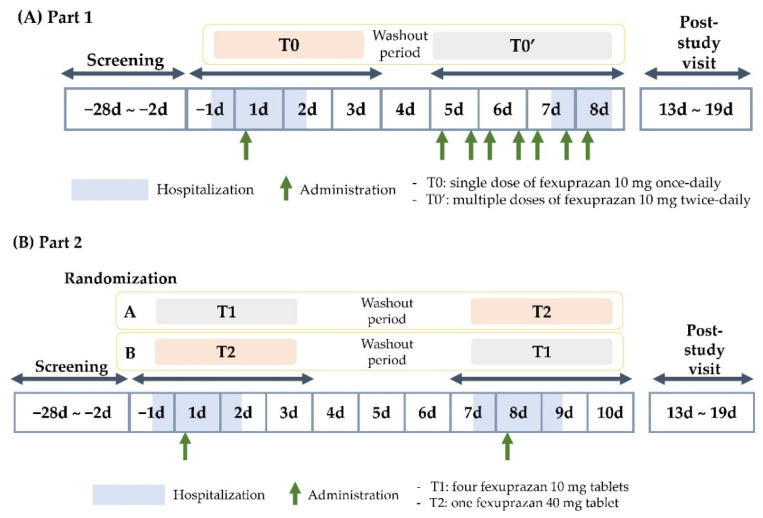
Study schema.

**Table 1 pharmaceuticals-16-01141-t001:** Comparison of the pharmacokinetic parameters of fexuprazan.

Part 1: Single Dose vs. Multiple Doses		
**Parameter**	**Single Dose Daily ** ** ^1^ ** **(*n* = 8)**	**Two Doses Daily ** ** ^1^ ** **(*n* = 8)**
C_max_ or C_max,ss_ (ng/mL)	8.26 ± 2.96	13.85 ± 4.96
C_min,ss_ (ng/mL)		5.36 ± 2.07
T_max_ or T_max,ss_ (h)	2.25 (1.50–3.00)	2.00 (1.50–3.00)
AUC_0–12h_ or AUC_0–12h,ss_ (h∙ng/mL)	53.47 ± 20.83	109.73 ± 40.52
AUC_last_ (h∙ng/mL)	84.91 ± 36.82	
Accumulation ratio		2.11
**Part 2: Four 10 mg Tablets vs. One 40 mg Tablet**		
**Parameter**	**4 × fexuprazan 10 mg ^2^** **(*n* = 23)**	**1 × fexuprazan 40 mg ^2^** **(*n* = 23)**
C_max_ (ng/mL)	34.62 ± 15.44	33.87 ± 15.04
T_max_ (h)	2.50 (1.50–6.00)	3.00 (2.00–6.00)
AUC_last_ (h∙ng/mL)	463.19 ± 169.78	446.24 ± 152.94
AUC_inf_ (h∙ng/mL)	479.88 ± 175.31	459.82 ± 154.55
Vz/F (L)	1372.91 ± 697.78	1341.60 ± 601.42
t_1/2_ (h)	9.91 ± 2.00	9.44 ± 1.65
CL/F (L/h)	94.60 ± 36.74	96.81 ± 32.69

^1^ single dose daily, a single dose of fexuprazan 10 mg once daily; two doses daily, multiple doses of fexuprazan 10 mg twice daily. ^2^ 4 × fexuprazan 10 mg, single dose of four fexuprazan 10 mg tablets; 1 × fexuprazan 40 mg, single dose of one fexuprazan 40 mg tablet. All values are presented as arithmetic mean ± standard deviation (SD) except for T_max_, which is presented as median (minimum–maximum).

**Table 2 pharmaceuticals-16-01141-t002:** Bioequivalence comparison between four fexuprazan 10 mg tablets and one fexuprazan 40 mg tablet in Part 2.

Pharmacokinetic Parameters	No.	Geometric LS Mean		Geometric LS Mean Ratio (T1/T2)		IntraCV(%)
		T1 ^1^	T2 ^1^	Point Estimate	90% CI	
C_max_ (ng/mL)	23	31.46	30.57	1.0290	0.9352–1.1321	19.0
AUC_last_ (h∙ng/mL)	23	431.30	419.15	1.0290	0.9476–1.1174	16.3

^1^ T1, four fexuprazan 10 mg tablets; T2, one fexuprazan 40 mg tablet. LS, least squares; CV, coefficient of variance.

**Table 3 pharmaceuticals-16-01141-t003:** Treatment-emergent adverse events following oral administration of four fexuprazan 10 mg tablets or one fexuprazan 40 mg tablet in Part 2.

Adverse Events	4 × Fexuprazan 10 mg ^1^ (*n* = 24)	1 × Fexuprazan 40 mg ^1^ (*n* = 24)	Total (*n* = 24)
Participants with TEAEs	2 (8.3) (3)	4 (16.7) (5)	5 (20.8) (8)
Abdominal pain	1 (4.2) (1)	1 (4.2) (1)	2 (8.3) (2)
Diarrhea	1 (4.2) (1)	1 (4.2) (1)	1 (4.2) (2)
Haematochezia	1 (4.2) (1)		1 (4.2) (1)
Nausea		1 (4.2) (1)	1 (4.2) (1)
Oedema peripheral		1 (4.2) (1)	1 (4.2) (1)
Headache	1 (12.5) (1)	1 (4.2) (1)	1 (4.2) (2)

^1^ 4 × fexuprazan 10 mg, four fexuprazan 10 mg tablets; 1 × fexuprazan 40 mg, one fexuprazan 40 mg tablet. Values are presented as the number of participants (percentage of subject) (number of TEAEs). TEAEs are presented by preferred term (PT) according to the MedDRA version 24.0.

## Data Availability

The data are not publicly available due to confidentiality reasons.

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
