# Peer review of "A Comparative Pharmacokinetic Study of Fexuprazan 10 mg: Demonstrating Bioequivalence with the Reference Formulation and Evaluating Steady State"

_pharmaceuticals, 2023, doi:10.3390/ph16081141_

Round 1

Reviewer 1 Report

The authors reported pharmacokinetics study for Fexuprazan with different dosing regimens, the bioequivalence of two dosages was verified. This paper may be publishable upon how well the authors address the concerns listed below:

1.       Full name of P-CAB is needed, full name of “IP” should be provided.     

2.       Table 1. It is more accurate to use “single dose daily” and “two doses daily” in this table and cross the main text. For twice-dose daily, it is unclear when the subjects consume the medicines.

3.       As there are only 8 subjects participated the Part1 study, why n=8 for both single-dose and twice-dose in Table 1?

4.       Figure 1. Why not show the full 48-hour C-T curve for subjects taking twice-dose daily? The quality of the C-T curves in Figure 1 is low. I suggest increase picture resolution.

5.       The exact dosage regimens for both studies are not clear. It is recommended to use chart or plot to demonstrate when the subjects take the medicines.

6.       It is not explained why using concentration at 7 d 12, 8 d 0 h and 8 d 12 h to determine if the steady state has been reached.  Also, only using three data points to do regression is not statistically significant. Why not plot the C-T for 8 days?

Importantly, the manuscript was poorly written. Many sentences are largely duplicated. The readability of this manuscript needs to be significantly improved.  

Author Response

Comment 1: Full name of P-CAB is needed, full name of “IP” should be provided.

Response 1: Thank you for reviewing our manuscript and for your insightful and helpful comments. The definition of P-CAB (potassium-competitive acid blockers) is included on line 55 and that of "IP" (investigational product) is included on line 181.

Comment 2: Table 1. It is more accurate to use “single dose daily” and “two doses daily” in this table and cross the main text. For twice-dose daily, it is unclear when the subjects consume the medicines.

Response 2: We have accordingly used “two doses daily” to avoid confusion. Additionally, we added a more detailed description of the IP administration in Part 1 (including the approximate time of administration) to the main text as follows:

Lines 286–292: The participants were admitted to the clinical trial center a day before the first administration and were instructed to for at least 10 h prior. The next day (1 d), at around 9 am, the participants received a single dose of fexuprazan 10 mg (T0) with 150 mL of water, and blood samples (6 mL) were collected at various time points, including pre-dosing (0 h) and post-dosing (0.5, 1, 1.5, 2, 2.5, 3, 4, 6, 8, 12, 24, and 48 h).

Lines 324–325: Participants were admitted the day before the administration of the IPs and discharged the day after dosing.

Comment 3: As there are only 8 subjects participated the Part1 study, why n=8 for both single-dose and twice-dose in Table 1?

Response 3: Thank you for your question. Part 1 of this study aimed to explore the pharmacokinetic parameters after administration of the newly formulated fexuprazan 10 mg both once daily and twice daily at a steady state. Therefore, we selected the minimum required number of participants in Phase 1 clinical trials, that is 8 participants. In the first-in-human study of the IP (reference #20), 8 participants per dose group were planned, without statistical justification of sample size. We explain this in more detail in the revised manuscript as follows (lines 286–299):

“It involved eight participants and aimed to explore their PK characteristics after single- and multiple-dose fexuprazan 10 mg administration. The number of participants was determined based on the ascending dose cohort in the first clinical study on fexuprazan (20). The participants were admitted to the clinical trial center a day before the first administration and were instructed to for at least 10 h prior. The next day (1 d), at around 9 am, the participants received a single dose of fexuprazan 10 mg (T0) with 150 mL of water, and blood samples (6 mL) were collected at various time points, including pre-dosing (0 h) and post-dosing (0.5, 1, 1.5, 2, 2.5, 3, 4, 6, 8, 12, 24, and 48 h). After a 4-day washout period, the participants visited the clinical trial center and were administered multiple doses of fexuprazan 10 mg twice daily (T0’) with 150 mL of water at approximately 9 am and 9 pm for 4 days (D5–7: approximately 9 am and 9 pm; D8: approximately 9 am). Upon reaching steady-state (D8), blood sampling was conducted at the same time points as in Period 1, up to 12 h, to determine the area under the concentration-time curve (AUC) from time zero to the end of the dosing interval.”

Comment 4: Figure 1. Why not show the full 48-hour C-T curve for subjects taking twice-dose daily? The quality of the C-T curves in Figure 1 is low. I suggest increase picture resolution.

Response 4: Thank you for your important suggestion. We have accordingly enhanced the quality and resolution of the image. To obtain one of the primary outcomes of part 1, the pharmacokinetic analysis involved utilizing plasma concentration data from 0 to 12 hours at steady state (day 8) in Period 2. The data were used to calculate the area under the concentration-time curve (AUC) from time zero to the end of the dosing interval (12h), as described in lines 297–299:

“Upon reaching steady-state (D8), blood sampling was conducted at the same time points as in Period 1, up to 12 h, to determine the area under the concentration-time curve (AUC) from time zero to the end of the dosing interval.”

Comment 5: The exact dosage regimens for both studies are not clear. It is recommended to use chart or plot to demonstrate when the subjects take the medicines.

Response 5: As suggested by the reviewer, we have added Figure 3, illustrating the study timeline, including when participants received treatment.

Comment 6:  It is not explained why using concentration at 7 d 12, 8 d 0 h and 8 d 12 h to determine if the steady state has been reached.  Also, only using three data points to do regression is not statistically significant. Why not plot the C-T for 8 days?

Response 6: Thank you for your important comment.  Our assessment of the steady state was based on the EMA's “OVERVIEW OF COMMENTS RECEIVED ON DRAFT GUIDELINE ON THE INVESTIGATION OF BIOEQUIVALENCE CPMP/EWP/QWP/1401/98 REV.1”, which describes the following:

"Achievement of steady state can be evaluated by collecting pre-dose samples on the day before the PK assessment day and the PK assessment day. A specific statistical method to ensure that a steady-state has been reached is not considered necessary in bioequivalence studies. Descriptive data is sufficient".

Thus, in our study, we adopted the conventional method described in this report by the EMA. Based on the half-life (9 h) of fexuprazan in the FIH study, it was assumed that the steady-state was achieved on D7, and accordingly, plasma concentrations on D7 12h, D8 0h, and D8 12h were used to assess the steady state.

This is described in lines 376–379. Additionally, we modified the plot as per your suggestion.

Reviewer 2 Report

The manuscript has presented a study on bioequivalence study of generic Fexuprazan generic compared to innovator and the pharmacokinetics at steady state. The manuscript has a poor flow and sequence. The quality of the manuscript needs to be be improved before consideration for publication.

(1) Abstract can be further elaborated by stating a short problem statement and research gap. 

(2) Introduction - The authors have given a thorough background of gastric acid related disease and medication. However, the authors should also highlight the research gap and significance of this study to make this article more appealing to the audience.

(3) Objective of the study should be mentioned clearly in the introduction.

(4) The sequence of the presentation is very poor. After Introduction should be materials and methods, and not the results and discussion.

(5) The authors should discuss briefly the ethical aspects of the study: whether the subjects are given food or at feeding state, withdrawal and termination procedure, measures to take care of the safety of the subjects.

(6) How much blood sample was withdrawn from the subjects?

(7) Please provide a brief sample preparation method. Any extraction involves to extract the drug from the plasma sample?

(8) The statistical analysis is not well presented in the article.

(9) The quality of figure 1(a) and (b) needs to be improved.   

The quality of English language needs to be improved.

Author Response

The manuscript has presented a study on bioequivalence study of generic Fexuprazan generic compared to innovator and the pharmacokinetics at steady state. The manuscript has a poor flow and sequence. The quality of the manuscript needs to be improved before consideration for publication.

Comment 1: Abstract can be further elaborated by stating a short problem statement and research gap. 

Response 1: Thank you for reviewing our manuscript and for your insightful and helpful comments. We have accordingly expanded the abstract.

Comment 2: Introduction - The authors have given a thorough background of gastric acid related disease and medication. However, the authors should also highlight the research gap and significance of this study to make this article more appealing to the audience.

Response 2: Accordingly, we have added the following description of the rationale and aim of study lines 71–77:

“Although the effectiveness of fexuprazan 10 mg (twice daily) has been demonstrated in phase 3 clinical trials in patients with gastritis, data on the pharmacokinetics (PKs) of administering fexuprazan 10 mg twice daily at a 12-h interval are lacking. Therefore, further investigation is necessary to address this crucial information gap. Moreover, it is imperative to ensure the bioequivalence of the new formulation with the previously approved 40 mg formulation.”

Comment 3: Objective of the study should be mentioned clearly in the introduction.

Response 3: We revised the sentences on lines 78–80 to clearly state the aim of the study:

“The present study aimed to evaluate the PKs of the new, recently approved, fexuprazan 10 mg formulation after single and multiple administrations (Part 1) and its bioequivalence to the previously approved 40 mg formulation was investigated (Part 2).”

Comment 4: The sequence of the presentation is very poor. After Introduction should be materials and methods, and not the results and discussion.

Response 4: Our manuscript follows the structure required by the journal. As per the journal’s guidelines, the manuscript should be structured as follows: Introduction, Results, Discussion, Materials and Methods, and Conclusions (optional). This structure is also applied in the template provided by the journal. Therefore, we conformed to the journal’s formatting guidelines. Kindly refer to these guidelines at: https://www.mdpi.com/journal/pharmaceuticals/instructions#preparation.

Comment 5: The authors should discuss briefly the ethical aspects of the study: whether the subjects are given food or at feeding state, withdrawal and termination procedure, measures to take care of the safety of the subjects.

Response 5: We have accordingly briefly described the ethical aspects you mentioned in lines 324–334:

“Participants were admitted the day before the administration of the IPs and discharged the day after dosing. Throughout their hospital stay, all meals were provided, except for breakfast on the day of dosing, and participants were only allowed to consume the meals and beverages provided by the clinical trial center. The decision to participate in the clinical trial was entirely voluntary for the participants, and declining to participate, had no consequences. Furthermore, participants retained the right to withdraw from the trial at any point, without facing any disadvantages in their future medical care. During the duration of the clinical trial, the researchers prioritized the safety of the participants, taking all necessary precautions to promptly and appropriately minimize any potential adverse effects, if they were to occur. Insurance coverage was obtained to cover any injuries that participants might sustain during the clinical trial procedures.”

Comment 6: How much blood sample was withdrawn from the subjects?

Response 6: We collected 6 mL of blood at each time point. This description was added to lines 292 and line 305.

Comment 7: Please provide a brief sample preparation method. Any extraction involves to extract the drug from the plasma sample?

Response 7: We have accordingly added the sample preparation method to lines 319–323:

“Blood samples were promptly collected in heparinized tubes for each blood sampling time point, and the plasma was separated through centrifugation at 1,900 × g for 10 min at 4 °C. The collected plasma samples were then transferred to 2 Eppendorf tubes at a volume of approximately ≥ 1.0 mL and stored in a freezer at ≤ -70 ℃ until further analysis.”

Comment 8: The statistical analysis is not well presented in the article.

Response 8: We have accordingly added a section on steady-state assessment and included more details on the statistical analysis, as follows:

Lines 376–380: “In Part 1, the method used to determine whether the curve is flat involved stepwise testing for linear trend to assess the steady-state. This assessment utilized pre-dose samples obtained on the day before the PK assessment day and the PK assessment day (21). For the analysis, plasma concentrations at D7 12 h, D8 0 h, and D8 12 h were considered.”

Lines 387–390: “Numerical data, such as demographic characteristics and PK parameters between the two treatments, were compared using an independent t-test or the Wilcoxon rank-sum test. Categorical data, such as TEAE incidence, were compared using the chi-square or Fisher’s exact test.”

Comment 9: The quality of figure 1(a) and (b) needs to be improved.

Response 9: Thank you for pointing this out. We have accordingly improved the quality of the figure you mentioned.

Round 2

Reviewer 1 Report

Thank the authors for adequately addressing my concerns. 

I suggest the authors to continue polishing their English to improve the readability of this paper.

Reviewer 2 Report

The authors have addressed all my concern.

Only minor editing is required.